# When One’s Not Enough: Colony Pool-Seq Outperforms Individual-Based Methods for Assessing Introgression in *Apis mellifera mellifera*

**DOI:** 10.3390/insects14050421

**Published:** 2023-04-27

**Authors:** Victoria G. Buswell, Jonathan S. Ellis, J. Vanessa Huml, David Wragg, Mark W. Barnett, Andrew Brown, Mairi E. Knight

**Affiliations:** 1School of Biological and Marine Sciences, University of Plymouth, Drake Circus, Plymouth PL4 8AA, UK; 2Information and Computational Sciences, The James Hutton Institute, Dundee DD2 5DA, UK; 3The Roslin Institute and Royal (Dick) School of Veterinary Studies, University of Edinburgh, Easter Bush Campus, Roslin EH25 9RG, UK; 4Beebytes Analytics CIC, Roslin Innovation Centre, Easter Bush Campus, Roslin EH25 9RG, UK; 5B4, Newton Farm Metherell, Cornwall, Callington PL17 8DQ, UK

**Keywords:** introgression, colony, *Apis mellifera*, ABBA BABA, ADMIXTURE, RAD-seq, SNP array, pool-seq

## Abstract

**Simple Summary:**

The human management of honey bees (*Apis mellifera*) has led to a reduction in the range and integrity of native subspecies, as they are replaced by foreign subspecies with traits that are perceived to be more desirable. Introgression—the transfer of alleles (gene variants) between genetically distinct lineages via hybridisation and repeated backcrossing—can lead to the loss of combinations of alleles that have built up over time as a result of adaptation. Measuring introgression is important for assessing the genetic integrity of colonies for breeders and conservationists. However, there is no agreed upon approach for measuring introgression in honey bee colonies. Here, we compare two commonly applied statistical methods for estimating introgression in honey bees from both individual and colony-pooled genetic data: a clustering approach (ADMIXTURE) and an allele pattern approach (ABBA BABA). Overall, data from a single individual from a colony resulted in lower introgression estimates than those from pooled colony samples using ADMIXTURE. However, ABBA BABA consistently yielded lower introgression estimates than ADMIXTURE. This study highlights that sometimes one individual is not enough to assess colony-level introgression and future studies using pooled colony approaches should not be solely dependent on ADMIXTURE for introgression estimates.

**Abstract:**

The human management of honey bees (*Apis mellifera*) has resulted in the widespread introduction of subspecies outside of their native ranges. One well known example of this is *Apis mellifera mellifera*, native to Northern Europe, which has now been significantly introgressed by the introduction of C lineage honey bees. Introgression has consequences for species in terms of future adaptive potential and long-term viability. However, estimating introgression in colony-living haplodiploid species is challenging. Previous studies have estimated introgression using individual workers, individual drones, multiple drones, and pooled workers. Here, we compare introgression estimates via three genetic approaches: SNP array, individual RAD-seq, and pooled colony RAD-seq. We also compare two statistical approaches: a maximum likelihood cluster program (ADMIXTURE) and an incomplete lineage sorting model (ABBA BABA). Overall, individual approaches resulted in lower introgression estimates than pooled colonies when using ADMIXTURE. However, the pooled colony ABBA BABA approach resulted in generally lower introgression estimates than all three ADMIXTURE estimates. These results highlight that sometimes one individual is not enough to assess colony-level introgression, and future studies that do use colony pools should not be solely dependent on clustering programs for introgression estimates.

## 1. Introduction

Introgression is the transfer of alleles between genetically distinct lineages via hybridisation and repeated backcrossing [1]. It is known to have important consequences for species and populations in terms of long term viability [2,3]. In some cases, introgression augments genetic diversity [4,5] via the introduction of new alleles boosting the adaptive potential of the population or individual [3,4]. Conversely, it can disrupt adaptation via the breaking up of co-adapted gene complexes [2,6]. Hybridisation and introgression can also lead to genomic extinction, when hybrids or introgressed individuals replace both or one of the parental lineages and the intact parental genome no longer exists [7]. Introgression and hybridisation also have the potential to lead to outbreeding depression [8], where genetically distinct lineages or species interbreed and the fitness of their offspring is lower than those of either parent [9]. These two contrasting consequences of introgression for the species or lineages involved raises questions about the extent to which introgression can occur without breaking down local adaptation and when the adaptive benefit of increasing genetic variance outweighs this potential cost. Consequently, assessing and monitoring introgression is an important topic in evolutionary and conservation biology.

The Western honey bee, *Apis mellifera* (Linnaeus, 1758), is a predominantly human-managed eusocial insect that provides an interesting case study for introgression due in part to the number of lineages and subspecies present across its natural range. *A. mellifera* has been classified into potentially 6 lineages, A, M, C, O, Y, and U, with more than 30 subspecies divided among them [10,11,12]. Primarily, Europe contains two evolutionary lineages, lineage M and lineage C (though lineage A is sometimes described as being present in the south of Spain). The M lineage is generally described as containing three subspecies, *Apis mellifera mellifera* (Linnaeus, 1758), *Apis mellifera iberiensis* (Engel, 1999) and *Apis mellifera sinisxinyuan* (Chen, 2016) [10,13]. In lineage C there are 10 subspecies described, including two of the most commercially popular subspecies, *Apis mellifera ligustica* (Spinola, 1806) and *Apis mellifera carnica* (Pollmann, 1879) [10,13]. These subspecies show morphological [14,15,16], genetic [17,18], and behavioural differences [14,19] (although empirical behavioural data is still wanting).

The distribution and demography of *A. mellifera* populations in Europe are highly influenced by human management, and particularly the widespread trade of honey bee subspecies across Europe. This is driven by the perception among beekeepers that certain subspecies have more desirable traits. The popularity of *A. m. carnica* and *A. m. ligustica,* for instance, is largely due to their alleged higher productivity and docility. In some cases, this popularity has led to the loss of local honey bee subspecies. For example, *A. m. ligustica* and *A. m. carnica* have introgressed with the original native subspecies *A. m. mellifera* in many countries, such as England, Wales, France, Denmark [20,21], Germany [22,23], and Poland [24].

*A. m. mellifera* has seen a large reduction in its range across northern Europe [25,26], and in its remaining refuges is under continued threat from introgression. Introgression levels vary throughout Europe, and a few remaining areas where introgression remains low have been described, for example, in Ireland [27], the Inner Hebrides (Scotland) [20,28], the Netherlands [28], and Norway [20,28]. As a result, numerous studies have focused on assessing introgression of the C lineage into *A. m. mellifera* [18,20,27,28,29,30,31,32,33].

One of the issues, however, with measuring introgression in a eusocial organism such as honey bees is that queens are polyandrous, and so multiple patrilines exist within a colony [34]. Currently, there is no agreed upon “best approach”, making comparison and overall assessment across different geographical areas difficult. Two broad approaches for assessing colonies have been used in previous studies: assessment by screening one single individual or by pooling several individuals from the same colony (“pool-seq”) [20,28,35,36]. Traditionally, pool-seq has been used to sample a population of unrelated individuals and came about to reduce the cost of population genetic studies [37,38]. The use of pools in this present context is therefore distinctly different as here it is used to represent a colony as a composite genome, which can then go on to be used to investigate the collective colony genotype [34]. In this method, the pooled individuals are not independent samples from a population, but a group of siblings or half-siblings. To reiterate, this is particularly important in the context of honey bees, where one worker is not fully representative of a colony due to the multiple mating of the queen. Pooled colony approaches have been used in studies of both honey bees and ants in this context [34,39]. In honey bee studies, pooled colony methods have been used to examine the genetic basis of traits such as aggression and calmness [39,40], and parasite defence [40]. Some introgression studies have also used pools of workers [28,35,36], whilst others have been conducted using individual workers [31,41] or drones [18,20,32].

Only one previous study has directly compared introgression values resulting from pooled and individual genotyping, and this was on a small scale (specifically on the SNP iPLEX MassARRAY platform during the development of the C-lineage introgression assay [28]). The C lineage introgression assay was developed to measure introgression from C-lineage honey bees (*A. m. carnica* and *A. m. ligustica*) into *A. m. mellifera* honey bees [18,28], and resulted in rigorously tested sets of ancestry-informative markers. The assay was largely designed for individual analysis, but Henriques et al. [28] examined the effects of pooled DNA on the ancestry estimates using two methods: testing the sensitivity of the iPLEX MassARRAY system to pools containing different numbers of individuals, and by applying the assay to four colonies in both individual and pooled form and comparing the resulting introgression estimates. Introgression values were consistently lower in the colony pool than in the individual (see Henriques et al. [28] Appendix A). Henriques et al. [28] noted that their findings merited further investigation using a larger sample size. Currently, this has been the only empirical test of introgression level differences between a pooled colony and individual workers, and it was specific to the iPLEX assay. However, Regan et al. [36] compared population assignment results from pooled whole genome sequencing to simulated individuals by using the program ADMIXTURE [42] (which results in a number of clusters, K, and individual samples memberships to clusters, Q), and reported results that were consistent with the pooled genotypes at lower K values and deviated marginally at higher K values. These results imply that, at lower K values, ADMIXTURE can be effectively used on pooled samples.

Alongside the choice of how to best sample a colony, the choice of genotyping approach is also important. Common genotyping methods for studying introgression are mitochondrial sequencing [43,44], microsatellite analysis [45,46], restriction site-associated DNA sequencing (RAD-seq) [47,48], and whole genome sequencing (WGS) [49,50]. Single nucleotide polymorphism (SNP) information has been demonstrated to outperform other methods for estimating honey bee introgression [32,51]. The generation of SNP data can be achieved through a number of genotyping methods. “RAD-seq” is one of a group of methods sometimes referred to as ‘genotype-by-sequencing’ or “reduced representation” methods (these include, among others, exome capture or transcriptome sequencing). RAD-seq involves sequencing a subset of the genome using restriction enzymes and allows for the same loci to be targeted across many individuals or pools without prior knowledge of the genome [52]. Importantly, as RAD-seq samples a subset of the nuclear genome (2% to 25% depending on restriction enzymes chosen [53]), the per-sample cost of sequencing is lower compared to WGS. RAD-seq can therefore allow an increased number of samples to be sequenced [52]. The benefit of this is especially important in population genetics, as some statistics and comparative analysis rely on a larger number of individuals or samples to be sequenced [34,52]. While WGS costs have fallen greatly in the last 20 years, it is still expensive to screen large numbers of samples [34].

Given the extent of imports of non-native honey bee subspecies (largely *A. m. carnica* and *A. m. ligustica*) into the UK [54] and the possibility of introgression resulting in the loss of native subspecies, the aim of this study is to compare introgression measurements resulting from both SNP array and RAD-seq datasets from pooled colonies and individual workers in putative *A. m. mellifera* colonies from the south west of England.

## 2. Materials and Methods

### 2.1. Overview of Sequencing and Data

For a direct comparison of methods, DNA from the same individual workers was processed twice: first using the C lineage introgression SNP assay [28], and secondly using an individual RAD-seq method. The pooled approach (pooled colony RAD-seq) used 30 workers sampled from the same colonies used for the individual approaches. The resulting data were then assessed via two commonly used introgression estimation approaches, firstly via the ADMIXTURE programme [42] and secondly using an ABBA BABA (also known as D statistics) approach [55,56,57,58].

This study used both novel data and data downloaded from the sequence read archive (SRA; Table 1). Novel sequence data (Table 1) were generated from honey bees sampled from South West of England through pooled colony RAD-seq, individual RAD-seq, and SNP iPLEX genotyping. The data downloaded from SRA used other platforms and sampling approaches (Table 1). To account for this, different bioinformatic workflows were implemented to generate the data required for analysis (Figure 1) separately for (1) subspecies standards and South West RAD-seq data using pooled colony samples; (2) subspecies standards and South West RAD-seq data for individual samples; (3) a combination of these data with WGS outgroup data (from *A. cerana*) (this outgroup was required for the ABBA BABA introgression analysis (Table 1)). The subspecies standards generated using pooled colonies and individuals were necessary for assessing introgression via the pooled colony and individual South West samples. Throughout, the pooled colony samples are referred to as ‘colony’ samples and the individual worker samples as “individual” samples.

### 2.2. Sample Collection

Honey bees were sampled in the South West of England (50.2660° N, 5.0527° W) in the summer of 2019, from beekeepers who suspected they had native *A. m. mellifera* or near-native honey bees and do not import foreign stock. A total of 30 colonies were selected from the South West. Samples consisted of 35 workers from each colony. Additional subspecies samples were obtained from experimental apiaries and breeding programs for use as standards. These consisted of *A. m. mellifera* (total *n* = 28) sampled in Sweden (*n* = 11), Norway (*n* = 10), and Switzerland (*n* = 7); *A. m. ligustica* (total *n* = 15) from Italy (*n* = 5) and Sweden (*n* = 10); and *A. m. carnica* (total *n* = 26) from Germany (*n* = 7), Sweden (*n* = 9), and Norway (*n* = 10). All samples were collected by bee keepers. Workers of unknown age were randomly selected and preserved in 70% ethanol until processed.

### 2.3. Restriction Site Associated DNA Methods

The double digest restriction associate DNA sequencing (RAD-seq) library preparation was adapted from Peterson et al. [52], and both the individual and pooled colony process consisted of a DNA extraction, DNA double enzyme digestion, adaptor annealing, adaptor ligation, size selection, and a PCR library enrichment.

#### 2.3.1. DNA Extraction of Individual and Pooled Colonies

All extractions were performed using an ammonium acetate protocol [59]. Of the 35 honey bees sampled per colony, one was randomly selected for the individual approaches, and 30 were used for the pooled colony approach. The rest were kept for contingency in case of insufficient DNA yield or thorax weight. For individual extractions, DNA was extracted from a standardized weight of thorax. For colony extracts, every thorax was weighed, and all honey bees donated an equal amount of tissue to each extraction. Five thoraces were pooled per extraction and six extractions were performed per colony. This approach was used due to practicalities, such as maximum volume of liquid in the tubes. The pooled thoraces were placed in 250 µL of Digsol digestion buffer, with 25 µL of Proteinase K, and placed in a 55 °C oven overnight. After digestion, 250 µL of ammonium acetate (pH 7.5) was added and mixed by vortexing four times over a 25 min period. The samples were then centrifuged at maximum speed (13,200 rpm), and the supernatant aspirated into a clean tube, discarding the pellet. Two washing steps then followed, first adding 1000 µL of ice cold 100% ethanol, inverting several times to mix, then storing at −20 °C for 2 h. The samples were then again centrifuged at 13,200 rpm for ten minutes and the supernatant was discarded. For the second wash, the same procedure was repeated using 70% ethanol and without a freezing period. The pellet was then dried on a heat block at 70 °C, before being resuspended in 20 µL molecular grade water and incubated at room temperature for 2 h. All samples were then treated with 4 µL RNase (New England Biolabs, Ipswich, MA, USA) in each extraction and incubated for 2 h at 37 °C. The individual samples underwent the same protocol, with two minor variations. The individual thoraces were incubated with 12 µL of Proteinase K and treated with 2 µL of RNase. All samples were then kept at −20 °C for long term storage. All extractions were quantified using a Qubit Fluorometer^®^ 2.0 (Thermo Fisher Scientific). The pooled extractions were equimolarly pooled into a final volume of 1000 ng DNA representing each colony, and individual workers were represented by 10 ng of DNA.

#### 2.3.2. Enzymatic Digest

The extracted genomic DNA was cleaned up using High Prep PCR clean-up system (MAGBIO Genomics, Gaithersburg, MD, USA) and digested using two restriction enzymes (MluCI and MspI). Colony extracts were digested in a reaction volume of 50 µL. The reaction contained 1.5 µL MluCI and 1.5 µL MspI (restriction enzymes), 5 µL CutSmart Buffer (10×) (New England Biolabs), 2 µL of molecular free water, and 40 µL of 1000 ng pool of DNA. This was incubated at 37 °C for 3 h. Individual samples were digested in reaction volumes of 10 µL, containing 0.5 µL MluCl, 0.5 µL Mspl, 1 µL of cutsmart buffer (10×) (New England Biolabs), and the 8 µL of 10 ng DNA. All samples were then cleaned again using High Prep^TM^ PCR clean-up system (MAGBIO Genomics) and eluted off with 25 µL of molecular grade water.

#### 2.3.3. Adaptor Ligation

All samples then had adaptors ligated to the enzyme digested DNA in batches of 12 samples, with each sample being assigned one of the 12 uniquely barcoded adaptors. The batches were not mixed in terms of extraction approaches, i.e., the batches consisted of either solely colony samples or solely individual samples.

For the pooled colony samples, adaptors were ligated in a 40 µL reaction containing 0.4 µL rATP 100 mM (Promega, Madison, WI, USA), 2 µL assigned P1 adaptor for that sample (4 mM), 2 µL P2 (4 mM), 0.5 ligase (New England Biolabs), 4 µL ligation buffer (New England Biolabs), 25 µL of enzyme digested DNA, and 6.1 µL molecular grade water. The individual samples had adaptors ligated in a 40 µL reaction containing 0.4 µL rATP 100 mM (Promega), 1 µL assigned P1 adaptor for that sample (4 mM), 1 µL P2 (4 mM), 0.5 ligase (New England Biolabs), 4 µL ligation buffer (New England Biolabs), 25 µL of enzyme digested DNA, and 8.1µL molecular grade water. All reactions were placed into a thermocycler and incubated at 23 °C for 30 min and 65 °C for 10 min, afterwards decreasing in temperature at a rate of 2 °C every 1 min 30 s until 23 °C was reached.

Both the colony batches and the individual batches, consisting of the now barcoded samples, were each pooled into a single tube, where each sample within was identifiable by one of the 12 unique barcodes. These pooled batches were then cleaned up using High Prep PCR clean-up (MAGBIO Genomics) to remove any unligated or adaptor-adaptor ligated products. The pooled batches samples were eluted off at a volume of 30 µL.

#### 2.3.4. Fragment Size Selection

All batches were taken through a size selection process using a Pippin Prep (www.sagescience.com, accessed on 24 April 2023). The Pippin prep machine was set to elute off read lengths between 150 bp to 500 bp.

#### 2.3.5. PCR Amplification

PCR was performed on each batch separately using a Phusion PCR kit (Thermo Fisher Scientific) to enrich the library sequences, add flowcell annealing sequences (primers regions specific to the Illumina platform), and to multiplex indices to all fragments. Multiplexing the batches allows the entire library to be combined by assigning a uniquely indexed reverse primer to each batch to create a unique combination of index and P1 combinations. Both individual and colony batched PCR reactions were set up in volumes of 25 µL, each containing 1 µL of reverse primer, 0.25 µL phusion polymerase, 10 µL of pooled adaptor ligated size selected DNA, 5 µL of phusion buffer (5×), 0.5 µL dNTPs (10 mM), 1 µL forward primer, 0.75 µL DMSO, and 6.5 µL molecular grade water. The reactions were then placed into a thermocycler for 98 °C for 3 min, then 16 cycles at 98 °C for 1 min, 63.5 °C for 1 min 30 s, and 72 °C for 1 min; the cycles were followed by a final extension of 72 °C for 3 min and an infinity hold at 4 °C. After PCR, the entire library was equimolarly pooled and quantified.

#### 2.3.6. Sequencing and Bioinformatics

Sequencing of the prepared library was performed by the Beijing Genomic Institute (BGI Hong Kong) on a Novaseq platform (Illumina, San Diego, CA, USA), using 150 bp paired end reads.

Raw reads were de-multiplexed in Stacks 2.48 with basic quality filters [60]. Then, using Trimmomatic (version 0.39) [61], reads were trimmed at the ends if quality dropped below a quality score of 4, a 4 base pair (bp) sliding window trimmed sections if the average quality dropped below 15 [61], and unpaired reads were discarded. Paired reads were mapped to the *Apis mellifera* reference genome (Amel_HAv3.1 assembly) using the Burrows–Wheeler aligner (BWA) MEM aligner (version 0.7.17) [62]. Reads were discarded if they aligned to more than one position on the reference genome. The subsequent alignment files were converted to the bam file format [63]. Samtools (version 1.10) was then used to filter reads with a mapping quality score >20, and the files were sorted by genomic coordinates and indexed [63]. Each sample bam file was edited to create one read group per sample (PICARD via GATK 4.2.0.0). The data were then processed using the Genome Analysis Tool Kit’s (GATK, version 4.2.0.0) best practices pipeline [64] (Appendix A). In GATK, the no ploidy setting was specified following Inbar et al. [34].

This was followed by an iterative filtering method performed in vcftools (version 0.1.16) to prevent erroneous SNP calls [65,66]. This robust filtering mitigates errors in downstream data analysis that can be caused by allelic dropout [65]. As well as quality filtering, SNPs were first filtered for depth and filtered based on their proportion of representation across all samples, and samples were filtered based on the proportion of all SNPs they contained. (Appendix A).

### 2.4. SNP Array Data Generation

In every case, the same individual was genotyped by the SNP assay and the individual RAD-seq. After DNA was extracted for the individual RAD-seq, the remainder of the honey bee tissue was sent to the Roslin Institute (Edinburgh, Scotland), who performed DNA extraction and library preparation for the SNP array platform. The SNP assay is accompanied with standards data from the Henriques et al. [28] assay. The standards supplied contain samples of *A. m. mellifera, A. m. carnica*, and *A. m. ligustica* honey bees.

#### SNP Array Variant Calling

SNP array raw data were formatted using a custom Python code into plink ped and map format for missingness filtering [67]. Genotypes were filtered in Plink (version 1.07) to obtain a genotyping rate of 0.9 using the “–geno” command [32,68]. Samples were filtered to contain a proportion of 0.9 of all SNPs using the “–mind” command. Data were then converted to the binary bed format using the “–make-bed” command in plink.

### 2.5. Subspecies Standards and Outgroup Data

#### 2.5.1. Individual Subspecies Standards

Subspecies standards were obtained for comparison with the individual RAD-seq samples. Whole genome data from Wallberg et al. [69] was downloaded from the Sequence Read Archive (SRA, project number PRJNA236426) using the sratoolkit (version 2.11.1). The downloaded data were from *A. m. mellifera* (*n* = 20) originating from Sweden and Norway, *A. m. ligustica* (*n* = 10) from Italy, and *A. m. carnica* (*n* = 9) from Austria (Appendix A).

These data were generated using a SOLiD 5500xl platform (Life Technologies, Thermo Fisher Scientific), which generates colorspace data. Due to the format of the raw data from this sequencing platform, a different bioinformatics procedure was required. Samples were run across multiple lanes and consisted of multiple colorspace fasta files per sample. A colorspace reference was constructed from the reference genome Amel_HAv3.1 assembly using bowtie’s ‘build’ command (version 1.2.3) [70]. The reads were then aligned to the colorspace reference using bowtie, generating output in the sam format [63]. Samtools (version 1.10) was then used to convert the reads to the bam format, merge the multiple bam files into their corresponding biological samples, and then sort and index the reads [63]. Read groups were edited using PICARD (via GATK 4.2.0.0) and bcftools (version 1.8), resulting in an mpileup file and a vcf file [71]. Colorspace data is prone to higher error rates than Illumina platform sequencing. Cridland et al. [72] examined this data set and found that the elevated error rate was associated with an excessive number of calls for triallelic sites, when compared to background rates of triallelic calls in *Drosophila* genomes. They also found that higher sequence coverage was associated with this higher error rate, when compared to an Illumina dataset. To control for this, triallelic sites were removed by filtering the data to contain only biallelic sites, and none of the high coverage sequenced samples were used. Additionally, data were filtered for genotype missingness (>0.9) and sample missingness (>0.9). The resulting vcf was finally filtered to contain only sites that were present in the individual RAD-seq data. Using bcftools, the data set was merged with the individual RAD-seq data, ready for downstream analysis.

#### 2.5.2. Pooled Colony Subspecies Standards

Standards required for the analysis of the pooled colony data were obtained from whole genome sequencing. This data set consisted of *A. m. mellifera* (total *n* = 28) sampled in Sweden (*n* = 11), Norway (*n* = 10), and Switzerland (*n* = 7); *A. m. ligustica* (total *n* = 15) from Italy (*n* = 5) and Sweden (*n* = 10); and *A. m. carnica* (total *n* = 26) from Germany (*n* = 7), Sweden (*n* = 9), and Norway (*n* = 10).

DNA extraction was performed using the same protocol as for the pooled RAD-seq data. Library prep and sequencing was performed at the BGI (Hong Kong, China) and sequenced on the BGISEQ-500 platform (BGI, Beijing, China) using 100 bp paired end sequencing. The raw data were received from BGI in fastq format.

Trimmomatic [61] was used to remove the ends of reads if they dropped below a threshold quality of 4. Reads were removed if they were below a length of 50 bp or had no paired read. A sliding window of 4 bp trimmed reads was applied if the average quality of the window dropped below a phred score of 20.

Reads were mapped to the *A. mellifera* reference genome (Amel_HAv3.1 assembly) using the (BWA) MEM aligner (version 0.7.17) [73], and discarded if they aligned to more than one position on the reference genome. The subsequent alignment files were converted to the bam file format [63]. Samtools was then used to filter reads with a mapping quality score >20, and the files were sorted and indexed [63]. Sample read groups were edited with PICARD (via GATK 4.2.0.0). The data were then processed using the Genome Analysis Tool Kit’s (GATK, version 4.2.0.0) best practices pipeline [64] (Appendix A). The data were then filtered for excessive depth. The mean depth (I.depth) was visualised in R (version 4.1.1), and sites were removed at a cut-off point of double the mean depth (minimum depth 5, maximum depth 65) [66,67]. The data were then filtered for missingness. SNPs were required to be represented in 0.9 proportion of all samples (vcftools max-missing), and each sample was required to contain a proportion of 0.9 of all SNPs in the data set (vcftools missing-indv) [66]. Finally, the data were filtered for sites that are present in the RAD-seq data and merged using bcftools [71]. The estimation of pool-colony allele frequencies was performed using methods from Inbar et al. [34]. Frequencies were estimated from the supporting reads for the reference and alternate allele in the AD (allelic depth) and DP (read depth) fields of the vcf file.

#### 2.5.3. Outgroup Data

The outgroup chosen for the ABBA BABA analysis was a separate *Apis* species, *A. cerana*, the eastern or Asian honey bee, as used in previous studies [72,74]. Paired-end whole genome data from 30 *A. cerana* worker bee samples generated by Chen et al. [74] on an Illumina HiSeq platform (Appendix A) were downloaded using the sratoolkit (Project accession PRJNA418874).

Here, the bioinformatics pipeline was identical to the pooled colony subspecies standards. Importantly, the reads were aligned to the *A. mellifera* genome for direct comparison of sites; therefore, not all reads align, and this limited the sites represented across the data sets. The resulting vcf file was filtered in the same manner as the pooled colony subspecies standards whole genome data, and then filtered to contain the RAD-seq sites. Two files were created, one for the individual and one for the pooled data. The vcf file that contained the individual RAD-seq sites was then merged with the individual data. The *A. cerana* vcf file that contained the pooled colony RAD-seq sites was used to generate the population allele frequencies for every site representing all 30 *A. cerana* individuals [66]. These allele frequencies were subsequently used alongside colony-level allele frequencies to perform the ABBA BABA analysis.

### 2.6. Introgression Estimators

Two methods were used to assess introgression: ancestry clustering using ADMIXTURE [42], and a genome wide test for introgression based on incomplete lineage sorting using Patterson’s D statistic and *f* statistic, also known as an ABBA BABA test [55,56,57,58]. ADMIXTURE uses genotypes and ABBA BABA uses allele frequencies.

#### 2.6.1. ADMIXTURE as an Introgression Estimator

ADXMITURE [42] (version 1.3.0) is a clustering algorithm that uses a maximum-likelihood model to estimate sample ancestry. ADMIXTURE estimates ancestry membership proportions (Q values) of individuals to clusters that represent ancestral populations (K). ADMIXTURE uses a cross-validation (CV) procedure to inform the most likely K value for the data. The most likely K value will exhibit a lower CV value than other K values [42]. For this program to be used for introgression estimation, standards are included in the analysis to view the clusters formed and the membership of samples to these clusters. ADMIXTURE was run using default termination criterion [75].

#### 2.6.2. ABBA BABA as an Introgression Estimator

The ABBA BABA approach involves fitting four populations onto a phylogenetic tree, and is based on examining derived and ancestral allele patterns brought about by incomplete lineage sorting (ILS) versus patterns of gene-flow and introgression [55,56,57,58]. ILS occurs when species or lineages undergo diversification into separate groups but there has been insufficient time for complete genetic differentiation of those groups. As a result, the gene tree differs from the overarching lineage or species tree, as the alleles are not perfectly segregated into those diversified groups [76,77].

ABBA BABA estimates three related statistics: Paterson’s D, the admixture proportion estimator *f*, and the *f*4-ratio. Paterson’s D examines deviations from the expected patterns of alleles resulting from ILS. The D statistic compares SNPs across the genome between three in-group populations (P1, P2, P3) and one outgroup population (Po) that match ABBA and BABA genotype patterns (Figure 2). An ABBA pattern is where population P1 has the ancestral allele (represented by ‘A’), while P2 and P3 share a derived allele (represented by ‘B’) (Figure 2). A BABA pattern is when P2 has the ancestral allele and P1 and P3 share the derived allele. Counting the occurrences of these patterns across all sites allows the investigation of whether the total number of shared derived alleles between two populations is greater than expected by chance. Effectively, the D statistic ascertains if P1 and P3 share an excess of derived alleles or if P2 and P3 share an excess of derived alleles (Appendix A). The related *f* statistics (*f* and the *f*4-ratio) estimate the overall proportion of admixture by comparing the excess of ABBA over BABA patterns to a scenario of complete admixture (Figure 3, Appendix A). This is performed by substituting P2 (Figure 3a) for another P3 population (Figure 3b) (for calculations see Appendix A and references [55,56,57,58,78,79,80]).

The *f*4 ratio is another method for estimating the admixture proportion, and results in a ratio of admixture proportions (α: 1-α) for the population being tested. Here, α-1 is the proportion of admixture from P3 into the admixed population, and α the proportion of P2 in the admixed population (Appendix A and [57,58,80]).

The pooled colonies were examined using Paterson’s D and *f*, calculated in a custom Python code using colony-level allele frequencies, and block jackknifing was performed using 20 blocks each containing 525 SNPs. To remain consistent within the ABBA BABA statistic, one colony each was chosen to represent the P1, P2, and P3 populations. This meant that the statistic was performed with either an *A. m. ligustica* or *A. m. carnica* colony sample in the P3 position, as opposed to a grouped C lineage sample. To rigorously estimate introgression, each pooled colony was tested multiple times using different colonies representing the subspecies standard at the P1 (*A. m. mellifera*) and P3 (*A. m carnica* or *A. m. ligustica*). Colonies were tested using combinations of 6 *A. m. mellifera* colonies, 6 *A. m. carnica* colonies, and 6 *A. m. ligustica* colonies, resulting in a total of 72 tests per South West colony (36 tests for the trio Mel; SW; Car, and 36 tests for Mel; SW; Lig). The selection of colonies to represent the subspecies was informed by the ADMIXTURE analysis, and only colonies that showed a 0.99 Q value to the subspecies cluster to which they putatively belonged were chosen (samples with low or no introgression).

The ABBA BABA approach requires allele frequencies; therefore, the individual dataset was examined using population allele frequencies, resulting in a population estimate of introgression. Paterson’s D and the *f* 4-ratio for the individual data set were calculated in Dsuite, using the individual data vcf file as an input [80].

## 3. Results

### 3.1. ADMIXTURE

The SNP array data consisted of 80 SNPs across a total of 197 samples, with a genotyping rate of 0.98. ADMIXTURE analysis identified the most likely K value as 2. This is unsurprising, as the ancestry informative SNPs chosen for the SNP array are designed to distinguish between the two *A. mellifera* lineages, C and M. To examine the accuracy of the SNP array Q values, the standards that accompany the SNP array were compared back to the original 117 SNP panel Q values in Henriques et al. [28]. There was a very strong correlation between the K = 2 Q value results and the Q values presented in the 117 SNP results in Henriques et al. [28] (R^2^ = 0.9981, Appendix A).

The individual RAD-seq data consisted of 23,916 SNPs across 61 individuals. ADMIXTURE identified the lowest CV value at K = 2. The pooled colony RAD-seq data consisted of 158,496 SNPS and 103 samples. ADMIXTURE analysis suggested K = 3 as the most likely number of clusters. These three clusters broadly represent the English South West samples, the *A. m. mellifera* samples, and the C-lineage bees, *A. m. carnica* and *A. m. ligustica*. The South West of England is unlikely to harbour its own ‘new’ subspecies, and the K = 3 result could be a result of a unique signature of admixture in this region. It is important to use biological knowledge of systems when interpreting ADMIXTURE results [81] and in order to compare the pooled results to the individual RAD-seq and SNP array results; here, K = 2 in the pooled RAD-seq is examined.

All South West samples in all methods showed some degree of introgression (Figure 4A–D, Appendix A). After filtering and quality control, there were 17 samples common to all three methods (SNP, individual RAD-seq and pooled RAD-seq, Figure 4 and Figure 5 and Table 2). Across these 17 samples, and across all methods, ancestry membership Q values for the M lineage ranged from 0.89 to 0.18 (Table 2 and Figure 4D), and for the C lineage from 0.82 to 0.11 (Table 2). Overall, the individual RAD-seq results and the SNP array results were highly correlated (Figure 4D), although pooled colony RAD-seq produced overall higher introgression values than the individual data (Figure 4D, Table 2). Sample c22 obtained the highest C lineage Q value in all three methods (Table 2 and Figure 5), and sample c7 had the lowest C lineage assignment in all three methods (Table 2), though in the pooled colony RAD-seq sample c25 was also assigned the same M lineage value (Table 2 and Figure 5). There were 4 samples where the individual RAD-seq and SNP array Q values yielded the same results (c8, c17, c23, c21, Table 2 and Figure 5). Average Q values for each ancestry cluster estimated across the 17 samples were the same for both the individual RAD-seq and the SNP array (M lineage 0.70 and C lineage 0.30, Table 2). The pooled colony RAD-seq generated a higher average C-lineage Q value assignment compared to the individual methods (0.38) and lower Q value assignment to the M lineage cluster (0.62) (Table 2). Some samples showed very similar Q values across all three methods, for example samples c21 and c22 (Table 2 and Figure 5). The largest sample Q value difference across methods was between pooled colony RAD-seq and the SNP array in samples c11, c13 and c2 (Table 2 and Figure 5). 

### 3.2. ABBA BABA

ABBA BABA for the pooled colony RAD-seq was performed on 10,505 SNPs (sites at which the *A. cerana* outgroup is fixed). Each South West colony was tested against different combinations of subspecies standard colonies (all tests and sample information are available in Appendix A). In the trios Mel; SW; Car and Mel; SW; Lig, a total of 19 pooled colonies were significant for positive D values (*p*-values < 0.05, Appendix A) across all tests regardless of the colonies chosen to represent the standards at P1 (*A. m. mellifera*) and P3 positions (either *A. m. carnica* or *A. m. ligustica*). This indicates introgression from either *A. m. carncia* or *A. m. ligustica*. The highest proportion of admixture estimated in both the Mel; SW; Car and the Mel; SW; Lig trios was for sample c22, with an average admixture proportion of 0.682 for *A. m. carnica* introgression and 0.626 for *A. m. ligustica*, respectively (Figure 6). The lowest *f* proportion of admixture value was in sample c25, with proportions of 0.06 (Mel; SW; Car) and 0.07 (Mel; SW; Lig) (Figure 6), and it was not significant for introgression. The standard deviations of the *f* statistic were generally low (Figure 6). The admixture proportions from *A. m. carnica* and *A. m. ligustica* trios were similar. The largest within-sample difference in admixture proportion was in c22 (a difference of 0.07) (Figure 6). Of the 9 colonies that were not significant in all combinations, 4 were not significant for both *A. m. ligutsica* and *A. m. carnica*, while 5 were not significant only for *A. m. carnica* introgression. All colonies that were not significant for introgression had admixture proportions <0.1.

Dsuite compared 4832 SNPs in the individual RAD-seq data set. D values deviated significantly from zero (*p*-values < 0.05) on all three trios tested (Table 3). The highest Z-score and smallest *p*-value were seen in the Car; South West; Mel trio. *A. m. mellifera* proportions (1-α) in the individual RAD-seq data ranged from 0.66 to 0.759 across the three trios, while the C-lineage proportion (α) in the South West population ranged from 0.241 to 0.340.

### 3.3. Summary of Introgression Estimates

For a comparison of introgression estimates resulting from the different genotyping and statistical approaches, a combined view is presented for 29 of the 30 colonies that were sampled from the South West (Table 4). Samples that failed during sequencing or did not pass quality checks are listed here as “no sample”, as they were not taken forward to analysis. Sample c12 failed completely and is therefore excluded from this table.

The ABBA BABA analysis yielded generally lower estimates of introgression for the pooled colonies than ADMIXTURE, with the exception of 4 samples (c7, c11, c15 and c19, Table 4, Appendix A). Some colonies produced disparate values from the ADMIXTURE and ABBA BABA approaches. For example, for colonies c4, c5 and c6, pooled colony ADMIXTURE Q values were 0.272, 0.348, and 0.257, respectively (Table 4), while the ABBA BABA approach estimated introgression values (*f*) of 0.09–0.08, 0.06–0.08, and 0.07, respectively, with neither c4 or c5 being significantly introgressed for *A. m. ligustica* introgression, and c6 not significant for either *A. m. ligustica* or *A. m. carnica*. A similar pattern was observed in colony c9 (ADMIXTURE SNP array Q value 0.22, colony RAD-seq Q value 0.24, ABBA BABA *f* proportion of 0.08 for *A. m. ligiustica* introgression and was not significant for *A. m. carnica* introgression). In both the ABBA BABA and the ADMIXTURE analyses, colony c22 was observed to have the highest introgression level (Table 4). Importantly, there were also samples where the pooled colony ABBA BABA and the ADMIXTURE SNP array gave similar results (colony c1, c2, c3, c13, and c17; Table 2 and Table 4 and Figure 6).

## 4. Discussion

This study compared estimates of introgression from three methods: the C lineage introgression SNP array [18,28], individual RAD-seq, and pooled colony RAD-seq, using two statistical approaches. Both the ADMIXTURE and ABBA BABA approaches revealed introgression in the *A. m. mellifera* population in South West England. Using ADMIXTURE, the pooled colony samples exhibited, on average, higher introgression values (Q values) than individual RAD-seq and SNP array results. The individual RAD-seq and SNP array methods produced similar Q values. This consistent ADMIXTURE result between the genotyping approaches based on samples of individuals was expected, as the SNP array has been rigorously tested to produce results similar to that of whole genome data. A comparison between ADMIXTURE Q value memberships and ABBA-BABA estimates based on the individual RAD-seq data (grouped into a population for estimating introgression by this approach) resulted in very similar introgression estimates. On the whole, the pooled colony ABBA BABA f statistics gave lower introgression values than the individual ADMIXTURE analysis (Table 4).

ADMIXTURE is designed for unrelated individuals, rather than pooled samples from related individuals, but nevertheless it has been used on pooled data in other honey bee studies [28,35,36]. These previous studies have reported contrasting results with those observed here. For example, the observed general pattern of pooled colony samples resulting in higher ADMIXTURE values is at odds with Henriques et al. [28], who reported that using the SNP array on pools resulted in lower ADMIXTURE values than those from individual samples. However, Henriques et al. [28] used a slightly different approach; here, DNA was equimolarly pooled (rather than tissue pools where tissue was mixed but DNA donation from each was not standardised). The sequencing platform was the SNP array, which is not ideally designed for capturing information from pooled samples. Our results also differ from Regan et al. [36], who found that pooled and individual data sets yielded similar ADMIXTURE results. They compared pool-seq ADMIXTURE results to a dataset simulated for individuals, and found that, up to K = 3, the results were consistent with those compared to the pooled genotypes, but at higher K values results started to diverge.

Another important consideration when using ADMIXTURE for introgression studies is that the number and quality of standards included can have a potentially significant effect on the results. The number of standards used here in the RAD-seq individual analysis was 40, while in the pooled analysis there were 69 (as many standards that were available were used). This is important, as ADMIXTURE uses input from all samples to infer clusters and there are no guidelines on the number of standard samples that one should place into ADMIXTURE to calibrate the analysis [81]. Nevertheless, the SNP array and the individual RAD-seq still resulted in largely similar values, even though they used different subspecies standards and numbers of samples tested.

ADMIXTURE is a population structure clustering programme that uses genotype calls and population clustering inference to estimate introgression [81,82,83]. There have been concerns raised about the use and over-interpretation of these methods, since they are sensitive to the choice of marker, the level of genetic differentiation, and the amount of data used in the analysis [84,85]. Consequently, alternative approaches, such as ABBA BABA, have been developed. Although the ABBA BABA approach has been seen to outperform both ADMIXTURE and STRUCTURE for identifying hybrids [83], ABBA BABA approaches can nevertheless result in higher error rates in situations with high incomplete lineage sorting [83]. In particular, an excessive amount of ILS between populations can reduce both the detection and the accuracy of the proportions of ancestry estimates in ABBA BABA approaches [83]. However, where pooled data have been used, complementing clustering approaches (e.g. ADMIXTURE) with methods based on ILS (e.g., ABBA-BABA) that can be implemented with pooled data (e.g., in DSuite or Poolfstat [80,86]) is recommended. This may be especially true for sequencing colony pools of honey bees, since evidence suggests that the inclusion of related or inbred individuals does not bias estimates of D, *f*, and *f*-4 [87]. However, there are still considerations that need to be taken when using these statistics. Attention to the block jackknife procedure is particularly important, as blocks need to have similar weighting (i.e., number of SNPs per block) and this can be compromised when blocks are based on genomic distance if there are areas that have a low genotyping rate.

If pool-seq experiments still wish to use ADMIXTURE, inference of the queen’s genotype from the pool-seq data is an alternative [35,88]. However, if the aim of a study is to reconstruct the queen’s genotype, the sampling of drones would be a much better use of sequencing efforts. We also highlight that, in this approach, inference of the queen’s genotype does not fully make use of the patriline information gained from pooled colony worker sampling (discussed in more detail below).

Concerns have been raised about the accuracy of allele frequency estimates from pool-seq studies [89]. Inaccurate estimates can occur when allele frequency estimates are generated from non-equimolar pools, sequenced at low depth, and use small pool sizes [90,91]. Here, these effects were mitigated through tissue weighing, equimolar pooling, and sequencing at a sufficient depth (average read depth of 40 in colony-pool samples). Additionally, the number of individuals chosen to represent the colony and capture patrilines is important, and needs to balance the cost incurred by the extra sequencing depth required to cover higher number of individuals per pool against the total number of colonies the study aims to screen.

Because there is no standard method in use for colony pooling in honey bee introgression studies, it is important to consider the information that different sampling strategies convey. For example, some studies have used multiple sampling of drones [20,32,33]. Sampling drones is effectively a maternity test (drones develop from unfertilized eggs), and does not carry information about patrilines present within the colony. Thus, if the queen is *A. m. mellifera* but has mated with multiple C lineage drones, this test will not reveal that, but it will allow insight into the lineage of drones that will mate in the local area (which is a benefit to beekeepers who wish to practice drone flooding, whereby a colony is used to produce drones for the local queens to mate with). If information is required about queen rearing, a more appropriate method would be to assess multiple workers or the colony-level genotype, as patrilines in the colony will affect the lineage of queens produced. Essentially, there is a significant distinction difference between asking “is this queen *A. m. mellifera*?”, “is this colony *A. m. mellifera*?”, or “is this individual *A. m. mellifera*?”, and it is important that beekeepers and researchers alike understand this difference and apply the right test, depending on the questions being asked. Such considerations are especially important when results are used to make conservation decisions, e.g., when reported to beekeepers. We recognise that pool-seq is not as cost effective and is more time consuming than the individual SNP array approach. The SNP array is commercially available to beekeepers, a great benefit of which is to allow for continued assessment of honey bee stocks in conservation and breeding programs. In contrast, pool-seq approaches require additional laboratory and bioinformatic steps. It was not the goal of this study to develop a more commercial platform but assess approaches for investigating honey bee introgression at the colony-level in scientific studies.

It is important to note that many populations in Europe have not yet been surveyed. The majority of past research to date focused on managed populations, including those in reserves. The identification of introgression does not in itself give an indication of population health, although it does indicate the degree of taxonomic integrity; both important questions that are beyond the scope of the present study. Additional work is also still needed to examine the effects of introgression on honey bees more widely. In previous studies, an arbitrary cut off for classifying an *A. m. mellifera* individual as introgressed has been used, for example, an ADMIXTURE Q value > 0.05 [92], >0.1 [27,93]. What introgression threshold is appropriate relates to the role of introgression in the disruption of co-adapted gene complexes. Simply put, at what level of introgression does *A. m. mellifera* still possess the traits that define it as *A. m. mellifera*? Discussion around this issue has begun in the literature [33]. Given that honey bees have a relatively high recombination rate [94,95], it could be hypothesised that the breaking up of co-adapted alleles would happen more rapidly than in species with lower recombination rates, especially as studies have observed that introgression rates are affected by recombination rates [96,97]. Answering these broader questions will allow conservationists and beekeepers to make better informed decisions.

## 5. Conclusions

Overall, there are some consistencies and some inconsistencies when comparing the ADMIXTURE and ABBA BABA approaches. However, ADMIXTURE is not designed for pooled data and can result in inconsistent introgression estimates for pooled data when results are compared to the ABBA BABA approach, which is specifically designed for use with allele frequency estimates. It is recommended, when assessing introgression using colony pooled data, that studies do not to rely solely on clustering programmes, or that they consider a queen genotype reconstruction approach before applying those analyses (while acknowledging patriline information may be lost). Pooled and individual approaches also produced inconsistent results, which is not surprising given honey bee biology. What this study underlines, however, is the need for careful consideration of the specific question being asked to ensure the best sampling and analysis approaches are used; there is little rationale, for example, for using one non-reproductive individual for measuring colony-level introgression in a polyandrous system, as an individual is not representative of the colony. Pools of workers give information about subsequent generations that would be produced from a colony, as well as a snapshot of introgression for that colony that accounts for both matrilines and patrilines within the colony. Pools of drones, in contrast, allow investigation of the queen’s genotype and information about the paternal alleles that will spread from that colony into the surrounding population (but not of the full lineage of dispersing new queens).

## Figures and Tables

**Figure 1 insects-14-00421-f001:**
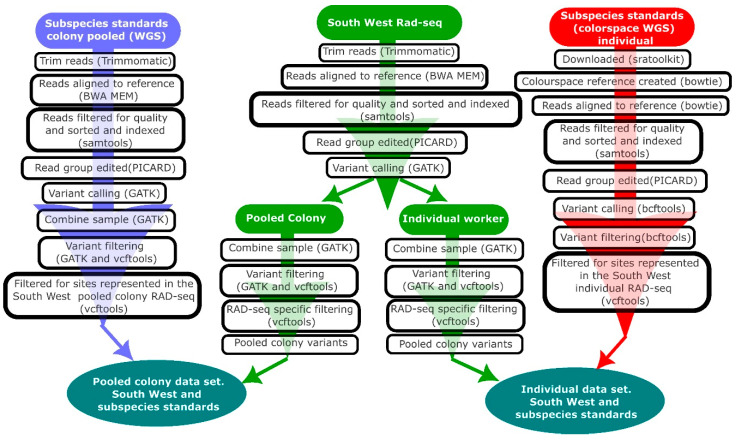
Overview of South West England RAD-seq samples and subspecies standards bioinformatics workflows. Samples were processed based on the sampling approach.

**Figure 2 insects-14-00421-f002:**
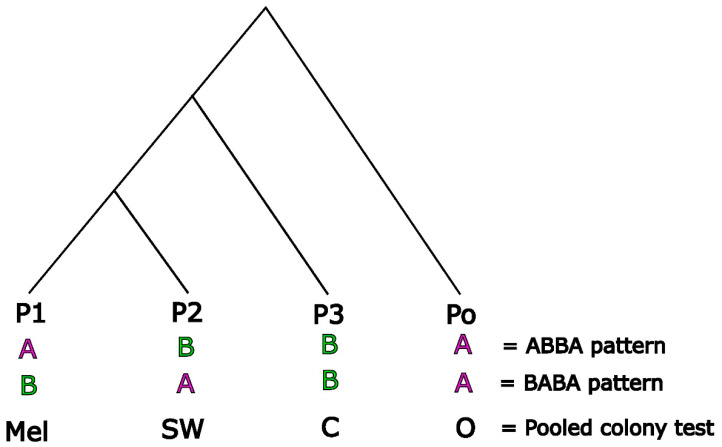
The principles of the ABBA BABA approach. Allele patterns of derived (“B”) and ancestral (“A”) alleles across the P1, P2, P3, and Po (outgroup) groups are illustrated. The ABBA pattern is when P2 and P3 share the derived allele, while P1 has the ancestral allele from the outgroup. The BABA pattern is when P1 and P3 share the derived allele, while P2 shares the ancestral allele with the outgroup. Here, pooled colonies were tested using *A. m. mellifera* as P1 (Mel), South West England as P2 (SW), and a c-lineage (C) colony in the P3 position, and the outgroup Po, represented by *A. cerana* (O).

**Figure 3 insects-14-00421-f003:**
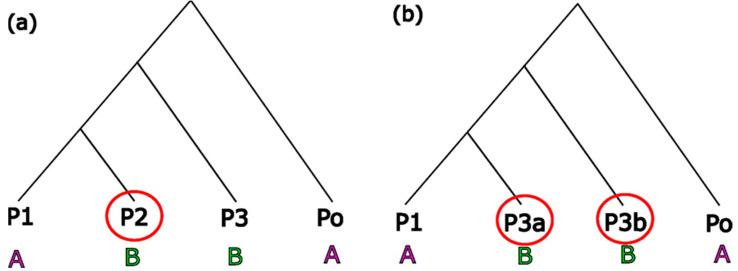
The *f* statistic compares the ABBA BABA counts to a scenario of complete admixture (**b**). Here, to simulate complete admixture, the P2 population (**a**) has been replaced with another P3 population, labelled P3a (**b**).

**Figure 4 insects-14-00421-f004:**
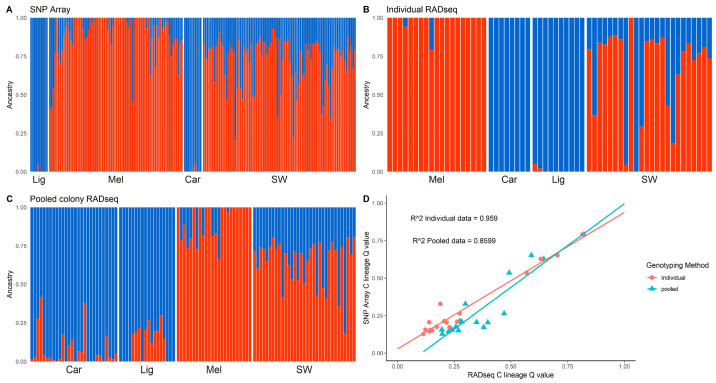
ADMIXTURE for K = 2 of honey bees in the South West of England (labelled SW) using (**A**) SNP array, (**B**) individual RAD-seq and (**C**) pooled colony RAD-seq. The blue cluster represents the C-lineage honey bees, *A. m. carnica* and *A. m. ligustica* (labelled Car and Lig, respectively), the red cluster represents *A. m. mellifera* (Mel). (**D**) Comparison of RAD-seq Q values to SNP array Q values for membership to the C lineage cluster, including an R^2^ value (denoted as R^2).

**Figure 5 insects-14-00421-f005:**
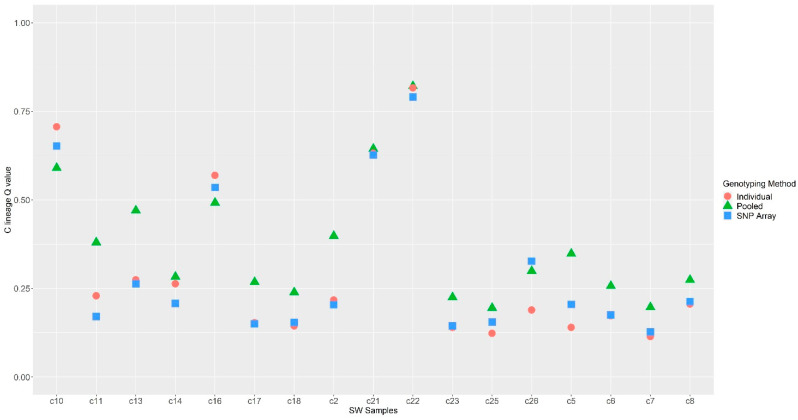
C-Lineage proportion estimated by ADMIXTURE at K = 2 using the SNP array (80 SNPs); individual ddRAD-seq (23,916 SNPs) and pooled colony RAD-seq (158,496 SNPs) across the 17 samples examined in all three methods.

**Figure 6 insects-14-00421-f006:**
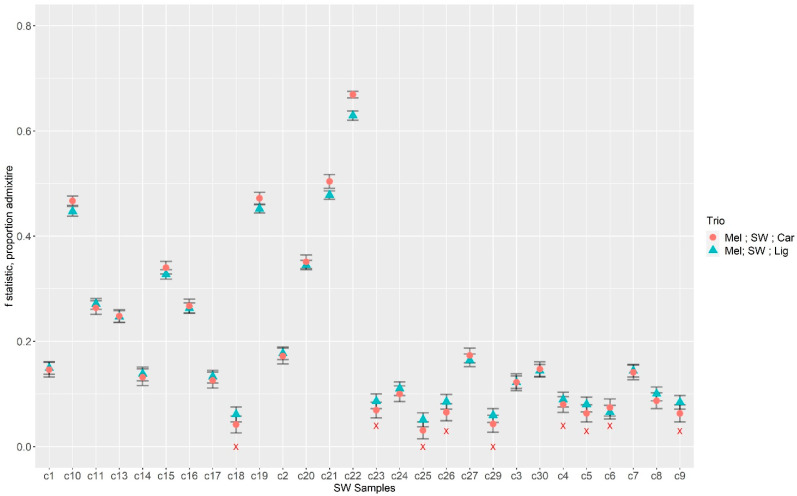
Average *f* statistics (proportion of admixture) calculated for pooled colonies. Trios tested for introgression between the South West and two different C-lineage subspecies. Shown here are trios consisting of *A. m. mellifera*; South West; *A. m. carnica* (Orange circle), and *A. m. mellifera*; South West; *A. m. ligustica* (blue triangle). A red X indicates colonies that had a non-significant result. Error bars represent the standard deviation of *f* for each South West colony estimate, calculated with every combination of *A. m. mellifera* and *A. m. carnica* or *A. m. ligustica* colony.

**Table 1 insects-14-00421-t001:** Overview of data sets used in this study. The samples represented three groups: the South West (representing the putative population of *A. m. mellifera* in the South West of England); subspecies standards, comprising three subspecies from across Europe (*A. m. carnica, A. m. ligustica* and *A. m. mellifera*); outgroup (representing the phylogenetic outgroup *A. cerana*). Genotyping methods used were restriction site associated DNA sequencing (RAD-seq), ancestry informative markers in the form of single nucleotide polymorphism (AIMs SNPs, [28]), and whole genome sequencing (WGS). The sequencing platforms used to generate the data were Illumina (Novaseq and HiSeq 2500, Illumina, San Diego, CA, USA), a MassARRAY iPLEX (Agenda Bioscience, Hamburg, Germany), a Beijing Genomics Institute (BGI) platform (BGI, Beijing, China), and a SOLiD platform (Thermo Fisher Scientific, Waltham, MA, USA) which generates colorspace data. Sampling approaches are either pooled colony, where 30 workers from the same colony are pooled for sequencing, or individual, where a single worker is sequenced.

Samples Representing	Genotyping Method	Sequencing Platform	Sampling Approach	Data Source
South West, England	RAD-seq	Illumina (Novaseq)	Pooled colony	Generated in this study
South West, England	RAD-seq	Illumina (Novaseq)	Individual worker	Generated in this study
South West, England	SNP Array	iPLEX MassARRAY	Individual worker	Generated in this study
Subspecies standards	WGS	BGISEQ-500	Pooled colony	Generated in this study
Subspecies standards	WGS	SOLiD 5500xl	Individual worker	Downloaded (SRA)
Outgroup	WGS	Illumina (HiSeq 2500)	Individual worker	Downloaded (SRA)

**Table 2 insects-14-00421-t002:** Q values of South West England samples that were included across all three genotyping methods. The ADMIXTURE Q values are membership values to two (K = 2) ancestral populations (clusters), here representing the C and M lineages from SNP array, individual RAD-seq and pooled colony RAD-seq.

Colony ID	ADMIXTURE K = 2					
	SNP Array		Individual RAD-Seq		Colony Pooled RAD-Seq	
	M	C	M	C	M	C
c2	0.80	0.20	0.78	0.22	0.60	0.40
c5	0.79	0.21	0.86	0.14	0.65	0.35
c6	0.82	0.18	0.83	0.17	0.74	0.26
c7	0.87	0.13	0.89	0.11	0.80	0.20
c8	0.79	0.21	0.79	0.21	0.73	0.27
c26	0.67	0.33	0.81	0.19	0.70	0.30
c11	0.83	0.17	0.77	0.23	0.62	0.38
c13	0.74	0.26	0.73	0.27	0.53	0.47
c14	0.79	0.21	0.74	0.26	0.72	0.28
c16	0.47	0.53	0.43	0.57	0.51	0.49
c17	0.85	0.15	0.85	0.15	0.73	0.27
c18	0.85	0.15	0.86	0.14	0.76	0.24
c10	0.35	0.65	0.29	0.71	0.41	0.59
c23	0.86	0.14	0.86	0.14	0.78	0.22
c21	0.37	0.63	0.37	0.63	0.36	0.64
c22	0.21	0.79	0.18	0.82	0.18	0.82
c25	0.84	0.16	0.88	0.12	0.80	0.20
Average	0.70	0.30	0.70	0.30	0.62	0.38

**Table 3 insects-14-00421-t003:** Dsuite results for the individual RAD-seq data examined on a population level. Populations are represented as *A. m. mellifera* (Mel), *A. m. carnica* (Car), *A. m. ligustica* (Lig), South West, and C-linage (*A. m. ligustica* and *A. m. carnica* combined in to one population).

Trios	D Statistic	Z Score	*p*-Value	F4-Ratio (α)	1-α
C-lineage; South West; Mel	0.105	3.47167	0.000517	0.292	0.708
Lig; South West; Mel	0.0816	2.63019	0.008534	0.241	0.759
Car; South West; Mel	0.1306	4.374	0.0000122	0.340	0.66

**Table 4 insects-14-00421-t004:** Summary of results from all sampling approaches and introgression estimates, where “nsi” indicates “no significant introgression”.

Sample ID	Summary of Introgression Estimates				
	ABBA BABA *f* Statistic Proportion of Admixture from C Lineage Colony		ADMIXTURE C Lineage Q Values from K = 2		
	Colony-Pooled Estimated Using *A. m. ligustica*	Colony-Pooled Estimated Using *A. m. carnica*	Colony-Pooled RAD-Seq	Individual RAD-Seq	Individual AIMs SNP Array
c1	0.149	0.146	0.277	no sample	0.189
c2	0.177	0.172	0.398	0.217	0.204
c3	0.122	0.122	0.258	no sample	0.137
c4	0.089	nsi	0.272	no sample	0.226
c5	0.080	nsi	0.348	0.140	0.205
c6	nsi	nsi	0.257	0.173	0.175
c7	0.144	0.141	0.197	0.114	0.127
c8	0.100	0.087	0.274	0.206	0.213
c9	0.084	nsi	0.239	no sample	0.216
c10	0.447	0.467	0.583	0.706	0.652
c11	0.271	0.264	0.380	0.229	0.171
c13	0.247	0.248	0.470	0.274	0.263
c14	0.138	0.132	0.283	0.263	0.208
c15	0.327	0.340	0.471	no sample	0.188
c16	0.263	0.267	0.492	0.569	0.535
c17	0.133	0.126	0.268	0.153	0.150
c18	nsi	nsi	0.239	0.144	0.154
c19	0.452	0.472	0.572	no sample	0.374
c20	0.345	0.351	0.525	no sample	0.434
c21	0.478	0.504	0.644	0.632	0.627
c22	0.629	0.669	0.821	0.816	0.790
c23	0.086	nsi	0.225	0.140	0.144
c24	0.110	0.100	0.305	no sample	0.259
c25	nsi	nsi	0.195	0.123	0.155
c26	0.085	nsi	0.299	0.368	0.327
c28	no sample	no sample	no sample	0.163	no sample
c29	nsi	nsi	0.257	0.168	no sample
c30	0.144	0.147	0.299	0.126	no sample

## Data Availability

Genetic data are available via the NCBI Short Read Archive, Bioproject: PRJNA921950. The custom python code used in this study is available at https://github.com/Vicbuz/When_ones_not_enough (accessed on 25 April 2023).

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
