# Peer review of "When One’s Not Enough: Colony Pool-Seq Outperforms Individual-Based Methods for Assessing Introgression in *Apis mellifera mellifera"

_insects, 2023, doi:10.3390/insects14050421_

Round 1

Reviewer 1 Report

The comments to author is attached as separate file.

Detailed comments can be found in the manuscript pdf file and supplementary file 1 & 2 

Reviewer 2 Report

The authors have suggested that a single individual form a colony does not accurately represent the introgression found at the colony level, while pooled sequencing better captures that variability. While I agree that one sample doesn’t represent an entire colony, the analyses made in the paper are not a 1:1 comparison. Thus, at times it is difficult to evaluate to what degree differences in admixture proportion estimates are truly because of pooled vs individual or may be a factor of data handling.

ADMIXTURE:

-       The ADMIXTURE results for the three methods, SNP array, individual RADSeq, and Pooled RADSeq are produced with different samples and different SNPs. While the authors take into consideration the variability in sampling by showing a comparison between 17 samples, it does not seem that the variability in SNPs is accounted for. We can expect that admixture proportions would be variable based on the number and variety of loci used (AIMS from the assay and randomly sequenced SNP from RAD). The authors should include, if possible, an analysis where the same loci are compared. If not, the authors should discuss the limitations of the results.  

-       Something to consider here is the genotyping of the pooled samples and whether these genotypes were treated as frequency of the pooled samples, or a consensus of the pooled samples. If the later, you risk drowning out low frequency variants. The authors also need to consider that alleles won’t be represented in equal portions of the pooled samples. A discussion on these limitations should be provided.

ABBA-BABA:

-       It appears that the ABBA BABA analysis produced lower admixture values for the pooled samples relative to the individual. Can the authors expanded on why the pooled colonies samples have a greater degree of variance among their sample results.

-       independently in the ABBA BABA analysis, and not just grouped together representing the ‘C lineage’.

-       Can the authors provide a visual/table comparing the ADMIXTURE and the ABBA BABA results so we can see a direct comparison.

Alternative comparisons:

-       Overall, the ADMIXTURE analysis with pooled samples resulted in higher levels of introgression relative to individuals. The results if the ABBA BABA analysis were lower relative to the ADMIXTURE analysis for pooled samples but there was no difference between analyses for individual samples. If you compared the ABBA BABA results for the pooled sample to the ADMIXTURE results for the individual samples, are the results similar? It seems like the pooled data coupled with the ADMIXTURE results may be the primary source of issue here. Since the authors have discussed that ADMIXTURE is not suitable for pooled data, but have also listed alternative methods (see below comment), are the proportions more consistent between alternative methods using pooled samples? Are these results then more consistent between the individual and pooled samples. This comparison would be ideal as it would more strongly highlight the concordance/disconsolance as result of methods or data type.

Methods:

Sampling:

-       It is difficult to parse information on sampling and which samples were used for which analyses. Can the authors provide a succinct table on the sample size for each category (Colony, M lineage and C lineage), data on which samples were used for each sequencing method, and data on which samples were used for each analysis type. 

Pooled SNP calling:

-       The authors used GATK to call SNPs on both individual and pooled sequenced samples. Can the authors please discuss the ploidy option used for SNP calling the pooled samples. The interpretation of genotype calls for pooled samples is not the same as individual samples. Genotypes would ideally by represented as a frequency of the n number of samples included in pooled sequencing. Ploidy specified will influence the genotype calculations. If no ploidy was stated (default diploid), can the authors show this does not have an effect on the genotypes.

-       Can the authors specify if frequencies or genotype calls were used for each analysis, how genotypes coded for pooled samples (frequency vs genotype) for each analysis (ie AMDIXTURE doesn’t used frequency data).

Alternative methods:

-       The authors mention tools specific for pooled samples in the discussion (Poolstat and Dsuites) which should be considered over ADMIXTURE. How do result from these tools compare to the results for ABBA BABA? It appears Dsuites was used for the individual samples, but not the pooled samples.

Sampling and analysis comparison:

-       It may be beneficial for the authors to consider a figure that outlines which comparisons are being made between analysis types and which samples are being used. It would just be easier to visualize the results.

Line 37: The authors should double check the inference methods of ADMIXTURE, which use maximum likelihood estimates, not Bayesian inference

Line 68: Other research has suggests there are more genetically distinct lineages (See Dogantzis et al 2021). Additionally, research from Cridland et al. 2017, has raised questions about the classification of the Z-lineage as potentially arising from signals of admixture.

Line 108: Please see Eynard et al. 2022 for a recent study comparing pooled and individual sequenced samples.

Author Response

Please see in the attachment.
